# ISIDORE, a Probe for In Situ Trace Metal Speciation Based on the Donnan Membrane Technique and Electrochemical Detection Part 2: Cd and Pb Measurements during the Accumulation Time of the Donnan Membrane Technique

**DOI:** 10.3390/molecules28020846

**Published:** 2023-01-14

**Authors:** Estelle Ricard, Jose-Paulo Pinheiro, Isabelle Le Hécho, Corinne Parat

**Affiliations:** 1IPREM, UMR 5254, Universite de Pau et des Pays de l’Adour, E2S UPPA, CNRS, 64053 Pau, France; 2Laboratoire Interdisciplinaire des Environnements Continentaux (LIEC), Unité Mixte de Recherche (UMR), Centre National de la Recherche Scientifique (CNRS), Université de Lorraine, UMR 7360, 54500 Vandœuvre-lès-Nancy, France

**Keywords:** Donnan membrane technique, screen-printed electrode, free ion concentrations, dynamic mode

## Abstract

The Donnan membrane technique (DMT), in which a synthetic or natural solution (the “donor”) is separated from a ligand-free solution (the “acceptor”) by a cation-exchange membrane, is a recognized technique for measuring the concentration of a free metal ion in situ, with coupling to electrochemical detection allowing for the quantification of the free metal ion directly on site. However, the use of the DMT requires waiting for the free metal ion equilibrium between the donor and the acceptor solution. In this paper, we investigated the possibility of using the kinetic information and showed that non-equilibrium experimental calibrations of Cd and Pb with the ISIDORE probe could be used to measure free metal concentrations under conditions of membrane-controlled diffusion transport. The application of this dynamic approach made it possible to successfully determine the concentration of free Cd in synthetic and natural river samples. Furthermore, it was found that the determination of free Cd from the slope was not affected by the Ca concentration ratio between the acceptor and donor solution, as opposed to the traditional approach based on Donnan equilibrium. This ISIDORE probe appears to be a promising tool for determining free metal ions in natural samples.

## 1. Introduction

The chemical forms of dissolved metals are mainly dependent on organic ligands that may strongly modify their bioavailability [1]. Therefore, different approaches have been proposed to determine metal speciation in natural samples. Some approaches involve the deployment of in situ exposure devices, such as a diffusive gradient in thin film (DGT) [2], permeation liquid membrane (PLM) [3], or the Donnan membrane technique (DMT) [4]. These devices, which accumulate metals after appropriate exposure in the field, are generally collected for measurement in the laboratory by inductively coupled plasma either with mass spectrometry (ICP-MS) or optical emission spectrometry (ICP-OES) detection. Other approaches involve the use of in situ measurements built on the hyphenation of a separation technique such as a gel-integrated microelectrode (GIME) [5,6] or a permeation liquid membrane (PLM) [7] hyphenated with an electrochemical detection system. Electrochemical techniques have also demonstrated their suitability for the in situ analysis of metal speciation with minimal sample preparation, reducing the risk of sample contamination and, consequently, the risk of speciation change [8]. Thus, direct in situ measurements such as the Absence of Gradient and Nernstian Equilibrium Stripping (AGNES) [9,10,11] or competitive ligand exchange-stripping voltammetry (CLE/SV) [12] were developed.

Regarding exposure techniques, although DGT techniques have gained a certain dominance among in situ exposure devices, the interpretation of their signal is not simple [13,14,15,16]. It was shown that relatively larger particles and/or humic matter could accumulate at the gel–water interface, which raises serious questions about the interpretation of the data [17]. PLM is subject to changes in transport flux in the solution between calibration and sample analysis that render the interpretation of dynamic information for colloidal metal complexes very difficult [18]. Direct speciation measurement with AGNES is an interesting and reliable method to determine the concentration of free metal ions in natural water such as freshwater; nonetheless, like all stripping techniques based on mercury electrodes, it is limited to a few metal cations [11].

Finally, DMT seems to be the best combination of signal interpretation, robustness, and ease of use of in in situ exposure techniques. The technique was developed and used for free metal ion measurement of cations such as Cd^2+^, Pb^2+^, Cu^2+^, Zn^2+^, and Eu^3+^ [19,20,21,22,23,24,25,26]. While the complexity of this equipment did not allow it to be transported to the field for in situ measurements in natural waters, Parat et al. (2015) overcame this problem by developing a new probe based on the hyphenation of a DMT device with a screen-printed sensor [27]. This probe has significant advantages over traditional DMT deployment, as detection can be performed in situ, directly in the acceptor solution, thus avoiding all the problems inherent in sampling, transport, and storage. This technique has been used to determine free Cd and Pb ions in natural samples such as freshwater rivers, and, due to the small volumes of the donor and acceptor solutions, equilibrium was reached in 6 h, much faster than the usual 36 h deployment. Although the transport of ions through ion exchange membranes has been extensively studied [4,28,29,30,31], no work has exploited this process in analytical applications. In this work, the dynamic information contained in the transient metal flux was considered and compared with the equilibrium situation.

Thus, we investigated the possibility of detecting Cd and Pb while they are accumulating in the acceptor solution (dynamic mode) rather than waiting for equilibrium to be reached in order to reduce the in situ deployment time. A calibration curve was performed during metal accumulation and compared to that obtained at Donnan equilibrium. Both calibrations were then compared to theoretical free Cd and Pb in synthetic samples, and the behaviour of metals in the membrane was discussed. Finally, the ISIDORE probe was tested in synthetic water doped in Cd and fulvic acids and in a river sample naturally contaminated with Cd.

## 2. Donnan Membrane Technique (DMT)

The ISIDORE probe is based on Donnan’s principle of membrane equilibrium [4]. Cations of the donor solution are transported across the membrane to the acceptor until the equilibrium of the Donnan membrane is reached. At equilibrium, the charge-corrected ionic activity ratios in the donor and acceptor solution are equal (Equation (1)):(1)$${\left(\frac{a_{i,donor}}{a_{i,acceptor}}\right)}^{\frac{1}{z_{i}}}={\left(\frac{a_{j,donor}}{a_{j,acceptor}}\right)}^{\frac{1}{z_{j}}},$$
where *a_i,donor_* is the activity of ion “*i*” in the donor solution in mol·L^−1^, *a_i,acceptor_* is the activity of ion “*i*” in the acceptor solution in mol·L^−1^, *a_j,donor_* is the activity of ion “*j*” in the donor solution in mol·L^−1^, *a_j,acceptor_* is the activity of ion “*j*” in the acceptor solution in mol·L^−1^, *z_i_* is the charge of ion “*i*”, and *z_j_* is the charge of ion “*j*”.

The main limitation of the first DMT devices was the long equilibration time of the order of a few days. Parat et al. proposed to enlarge the inner diameter of the rings to obtain a larger membrane surface area and to reduce the thickness of the central ring and, consequently, the volume of the acceptor solution [27]. Thus, these modifications reduced equilibration times by approximately 6 h for environmentally relevant concentrations of Cd and Pb. In addition, coupling the DMT cell to an electrochemical detection system allowed for the direct determination of the free ion concentration without having to bring the acceptor solution to the laboratory for ICP-MS analyses.

## 3. Results and Discussion

### 3.1. Non-Equilibrium ISIDORE Probe Calibration

The donor solution was simultaneously spiked with Cd and Pb. For each concentration, the experiments were performed until the Donnan equilibrium was reached in order to determine the plateau and the accumulation slope. Figure 1 shows that whatever the studied element, the accumulation curve across the membrane is classically broken down into four parts. First of all, time is required before an analytical signal is obtained in the acceptor solution. This behaviour is associated with the initial accumulation of metal in the membrane [27,29]. After this delay, a linear accumulation (slope) is obtained for both elements, followed by a non-linear region, before finally reaching a flat region or plateau that corresponds to the Donnan equilibrium [29].

The average plateau value was calculated from the average of all plateau points, while the slope value was calculated by keeping the points recording from 0 h 31 min to 2 h 34 min for Cd and from 1 h 16 min to 4 h 22 min for Pb, corresponding to the linear accumulation of free metals in the DMT (Figure 1). For both metals, good linearity with correlation factors larger than 0.99 was obtained during the accumulation phase, indicating that the flux of metal ions across the membrane is constant and opening up the possibility of using the slope as dynamic information (Figure 1).

This experiment was repeated for the four concentrations of Cd and Pb. For each concentration, the slope and plateau values were determined and plotted against the free ion concentration in the donor estimated by Visual Minteq (Figure 2). As expected, the metal ion flux depends on the concentration of free metal in the donor, with higher concentrations resulting in higher fluxes. Thus, for both metals, the slope value of accumulation increases linearly with the free ion concentration in the donor, suggesting that this value can be used for calibration.

### 3.2. Behaviour of Cd and Pb in the Membrane

The behaviour of the two metal ions is quite different since the plateau is reached after approximately 5 h for Cd but requires more than 10 h for Pb (Figure 1). In order to understand the difference in behaviour between Cd and Pb, we sought to determine whether the diffusion of the free metals across the membrane was controlled by solution diffusion or diffusion in the membrane. The charged membrane constitutes a Donnan phase where the metal ion concentrations will be higher than in the two adjacent solutions (donor and acceptor). Furthermore, as these solutions are kept well mixed by continuous circulation, the thicknesses of the inner and outer diffusion layers are fixed after a short initial transient time. Finally, as the electrolyte concentration is assumed to be equal in the acceptor and donor, there is no electrostatic potential gradient in the membrane, and ion transport between the solution and the membrane will be controlled by the diffusion layer at the membrane interface.

The problem can be formulated as two diffusion layers (one on the acceptor side and one on the donor side) separated by a negatively charged membrane that can be crossed by cationic species. From a mathematical point of view, this is a complex problem described by a set of differential equations varying in time and space. The two transport equations in the solution are similar, simplifying the problem somewhat, and can be further simplified if one type of transport is clearly dominant over the other. The two types of transport are called membrane-controlled or solution-controlled diffusion.

Weng et al. presented a detailed mathematical formulation of the problem and a detailed numerical solution as well as an approximated analytical solution to solution diffusion control (valid if the complexation in the donor and acceptor are equal) (Equation (2)) [30]:(2)$$\frac{C_{i,tot,\;acc}}{C_{i,\;tot,don}}=A_{1}\cdot t\;with\;A_{1}=\frac{A_{e}D_{i}}{\delta V_{acc}},$$
and membrane diffusion-controlled fluxes (Equation (3)):(3)$$\frac{C_{i,\;acc}}{C_{i,\;don}}=A_{2}\cdot t\;with\;A_{2}=\frac{A_{e}D_{i,m}B^{Zi}}{\delta_{m}V_{acc}P_{i}},$$
where *C_i,tot,acc_* and *C_i,tot,don_* are the total concentrations of ion *i* in the acceptor and donor solutions, respectively, while *C_i,ac_* and *C_i,don_* are the concentration of the free ion *i* in the acceptor and donor solutions, *t* is the time (s), *A*_e_ is the effective surface area of the membrane, *D_i_* is the diffusion coefficient of ion i in water, *D_i,m_* is the apparent diffusion coefficient of ion i in the membrane (*D_i,m_* = *D_i_*/*λ_i_* m^2^·s^−1^ with *λ_i_* the tortuosity factor for ion *i* in the membrane), *B* is the Boltzmann factor*, δ* is the thickness of diffusion layer in solution, *δ_m_* is the thickness of the membrane, *V_acc_* is the volume of the acceptor solution, and *P_i,ac_* and *P_i,don_* are the complexation factors in the acceptor and donor solutions.

In order to understand whether there is solution diffusion control or membrane diffusion control for our systems, we performed calibration curves for Cd and Pb. When the transport is controlled by solution diffusion (Equation (2)), all parameters are previously estimated, respectively, *D*_Cd_ (7.3 × 10^−10^ m^2^·s^−1^) and *D*_Pb_ = 8.1 × 10^−10^ m^2^·s^−1^) [32], *V_acc_* (7 × 10^−6^ m^2^), and *δ* = 1 × 10^−4^ m in our previous work [27], and finally *A_e_* = 2.74 × 10^−4^ m^2^ which corresponds to 20% of the membrane surface, as suggested by Weng 2005 [30]. Using these values, we obtained a prediction for the time to reach 95% of equilibrium concentration (t95%) of 2 h 55 min for Cd and 2 h 33 min for Pb, which is much shorter than the experimental results. Indeed, using these parameter values, Equation (2) does not provide a good fit to the experimental curves (Cd Figure 3A and Pb Figure 3C), suggesting that Cd and Pb transport are not controlled by solution diffusion.

Unlike Equation (2), there are three unknown parameters in Equation (3): the tortuosity factor *λ_i_*, the Boltzman factor *B*, and the degree of complexation in the acceptor *P_i_*. Weng et al. (2010) suggested a tortuosity factor of 60 for copper and calculated *B* values of 29 and 9.6 for Ca(NO_3_)_2_ concentrations of 2 mM and 20 mM, while the *P_i_* values depend on the acceptor solution composition [33]. In our case, the best estimate for *B* is a value of 13.7; however, this is a rough estimate as our electrolyte is a mixture of 3 mM Ca(NO_3_)_2_ and 5 mM acetate buffer for which the calculation of a Donnan factor is quite involved. The best description of the Cd and Pb transport data for the four concentrations was obtained for a *P_i,ac_* of 2.75 and 8 (Cd Figure 3B, Pb Figure 3D), leading to prediction times to reach equilibrium (t95%) at 4 h 22 min and 11 h 27 min, which are in good agreement with the experimentally determined times. According to Weng et al., there is no significant effect of the ligands on the equilibrium time when *P_i_* is relatively small (<50) [30]. Thus, it is clear that both Cd and Pb transport is controlled by diffusion in the membrane.

Membrane diffusion control provides a simple dynamic equation describing the accumulation of metal ions in the acceptor over time since only free metal ions contribute to transport and accumulation. For trace metal speciation, membrane diffusion control is preferable to solution diffusion control because of its simplicity since, for all practical purposes, the DMT functions as an ion-selective electrode, where only the free metal concentration is relevant during both calibration and sample measurements. On the other hand, if solution diffusion control is the control mechanism, Equation (2) must be modified to take into account the contribution of labile complexes to transport and the impact of the different diffusion coefficients of metal complexes in the transport and in the thickness of the diffusion layer as described by Domingos et al. for the Permeation Liquid Membranes (PLM) [18].

Although metal transport by diffusion in solution or in the membrane can be approximated by models, the fit is not perfect, especially for higher metal concentrations. This is probably due to an incomplete physico-chemical description of the membrane, namely the Donnan effect and its impact on transport across the membrane and the fact that in our device, the outer and inner geometries are quite different, which can impact the solution transport. A better theoretical description of this transport is therefore required, for example, to explain the differences between Cd and Pb ions.

### 3.3. Analysis of Synthetic and Natural Samples

Finally, the Isidore probe was applied to synthetic and natural Cd-doped samples. Figure 4 shows the free Cd concentrations determined from the plateau (Donnan equilibrium) and the slope (dynamic accumulation) for different samples. As expected, the calibration curve is close to the y = x line, which confirms the idea that it is possible to cut the measurement time in half using the slope instead of the plateau. However, the results of the river samples show differences in free Cd concentrations depending on the determination method, slope or plateau (Table 1). This behaviour is particularly marked for the synthetic river samples, which show an overestimation between 34 and 64% (Figure 5 red triangles). Table 1 shows that this overestimation of the calculated equilibrium concentration is not linked to the presence of fulvic acids. Therefore, this overestimation may involve the behaviour of Ca during the experiment. Indeed, during the calibration of the ISIDORE probe, the composition of the donor and acceptor solutions are identical (3 mM), and therefore the Ca concentration in the donor and acceptor solutions is in equilibrium. In the case of synthetic and natural waters, we noticed an imbalance in the Ca composition, with ratios [Ca^2+^]*_don_*/[Ca^2+^]*_acc_* between 0.4 and 0.7. Similar ratios are observed for natural samples, but the effects are probably attenuated due to the lower concentrations of free Cd.

In the DMT system, the presence of a multivalent cation in the background solution (usually, Ca^2+^) is necessary to compete with the target metal for binding to the membrane and to ensure sufficient transport of the target cation through the membrane [25]. However, according to Equation (1), the concentration of Cd measured in the acceptor solution at Donnan equilibrium is equal to that measured in the donor solution only when the concentration of Ca is the same on both sides [4]. The Cd concentration determined from the plateau was therefore corrected by the ratio [Ca^2+^]*_don_*/[Ca^2+^]*_acc_*. Figure 5 shows the Cd concentrations determined from the slope versus the Cd concentrations determined from the plateau after correction according to Equation (1). A good correlation is obtained between the two methods (r^2^ = 0.92) regardless of the type of samples. These results show that the Ca concentration is an important parameter to consider when determining the free metal concentration at equilibrium, but it does not seem to affect the accumulation rate of Cd in the acceptor solution. Determining the free ion metal during the accumulation time appears to be an interesting alternative to the Donnan equilibration method because this approach is both faster and not affected by the Ca concentrations in the donor and acceptor solutions.

## 4. Materials and Methods

### 4.1. Reagents

Stock solutions of Cd, Pb, and Hg at 1000 mg·L^−1^ were obtained from Merck. Potassium nitrate (KNO_3_—Trace Select), nitric acid (HNO_3_ 69–70%, Baker Instra-Analysed for trace metal analysis), sodium hydroxide (NaOH Baker Analysed), hydrochloric acid (HCl Baker Instra-Analysed for trace metal analysis), and ethylenediaminetetraacetic acid (EDTA) were purchased from Aldrich. Acetic acid (CH_3_COOH, Trace Select) and sodium acetate trihydrate (CH_3_COONa, 3H_2_O Trace select) were obtained from Fluka. Magnesium sulphate (MgSO_4_·7H_2_O, Pro Analysis), calcium chloride (CaCl_2_·2H_2_O, Pro analysis), and calcium nitrate (Ca(NO_3_)_2_) were purchased from Merck. Sodium hydrogen carbonate (NaHCO_3_) was obtained from Scharlau. Nordic Aquatic Fulvic acid Reference (1R105F) and Suwannee River Fulvic Acid Standard II (2S101F) were purchased from the International Humic Substances Society (Denver, CO, USA).

Ultra-pure milli-Q water (resistivity 18.2 MΩ cm) was employed in all the experiments. A stock solution of acetate buffer (0.1 M, pH 4.6) was prepared by mixing appropriate amounts of CH_3_COOH and CH_3_COONa. A stock solution of synthetic river water was prepared as follows: 10 mM NaHCO_3_, 1 mM MgSO_4_, 2 mM CaCl_2_, and 0.5 mM KNO_3_, which corresponds to an ionic strength of 19 mM and a pH of 7.5 ± 0.2.

The cation exchange membrane (VWR International, Radnor, PA, USA) used in this work has a matrix of polystyrene/divinylbenzene with sulphonic acid groups that are fully deprotonated above pH 2. The ion-exchange capacity is 0.8 meq g^−1^, the thickness of the membrane *δ_m_* is 1.6 × 10^−4^ m, and its surface area is 5.3 cm^2^ [4].

### 4.2. Equipment

Stripping Chronopotentiometry (SCP) measurements were performed with an Eco Chemie μ-Autolab III potentiostat controlled by the GPES 4.9 (Eco Chemie) software package. Temperature and pH were checked with a multi-parameter analyser (WTW 340i). The flow-cell (DropSens) was modified to fit the homemade screen-printed sensor (SPE). The SPE was prepared in the laboratory by means of a carbon commercial ink (Electrodag^®^ PF 407A) for the working and counter electrodes and an Ag/AgCl 3:2 ink (Electrodag^®^ 6037 SS) for the pseudo-reference electrode [27]. Prior to any electrochemical measurements, a thin layer of mercury was deposited onto the surface of the screen-printed electrode. The Hg deposition on the working surface area was carried out using an acetate buffer solution (0.1 M, pH = 4.6) doped in Hg at 0.83 mM. ICP-MS measurements were performed with an Agilent ICP-MS (7500 series, Agilent Technology, Santa Clara, CA, USA) and Ca concentration was measured by an iCAP 6000 series (Thermo Scientific™, Waltham, MA, USA).

### 4.3. ISIDORE Probe

The design of the ISIDORE probe measurement is explained in detail by Parat and Pinheiro [27]. Briefly, a DMT cell containing the acceptor solution is immersed in a donor solution, i.e., river water. The acceptor solution, in which the free metal ions present in the donor solution will accumulate, is connected to a flow-cell in which an SPE is placed. The circulation of the solution from the DMT to the measuring cell is achieved by means of a peristaltic pump (Labcraft Hydris 05) (Figure 6).

Before analysis, the DMT membranes were prepared by shaking them several times in succession with EDTA (0.1 M) to remove trace metal impurities, then 1 M Ca(NO_3_)_2_ and 3 mM Ca(NO_3_)_2_, which is the concentration of the background electrolyte solution used in the experiment. In the last step, the pH is controlled to ensure no more protons are released.

For all the experiments described below, the acceptor solution is composed of a mixture of 3 mM of Ca(NO_3_)_2_ solution and 5 mM acetate buffer in order to maintain a constant pH (4.6 in this study) during the electrodeposition of the metal on the surface of the electrode [34] and also to avoid overlap of Cd and Pb peaks [27].

The ISIDORE probe was calibrated for four different concentrations of Cd and Pb, corresponding to free Cd concentrations of 39, 78, 117, and 157 nM and free Pb concentrations of 14, 28, 43, and 56 nM. The assembly of a clean DMT and a new SPE was performed for each concentration. The background solution in both donor and acceptor solutions was 3 mM Ca(NO_3_)_2_ buffered at pH 4.6 with 5 mM of acetate buffer. The acetate buffer was used to keep the pH constant on both sides throughout the experiment. As there were no ligands in the acceptor solution, the concentration of free ions in the donor solution corresponds to that measured in the acceptor solution. The donor solution was kept under stirring throughout the experiment.

Electrochemical detection was carried out in two steps, a deposition step followed by a stripping step. In the first step, the deposition potential of −1.6 V was maintained for 120 s under a solution flow of 2 mL·min^−1^, after which the flow was stopped for 10 s and the potential maintained at −1.6 V. Then, in the stripping step, the potential signal was measured as a function of time under an applied stripping current of *I_s_* = 10 μA.

### 4.4. Natural Samples

Two freshwater rivers were used. The first river is the “Luy de Béarn”, a 76.6 km long French river that starts at Andoins (Pyrénées-Altantiques, southwestern France) and ends at Gaujacq, where the “Luy de Béarn” merges with the Adour. For this river, named here the “Uzein River”, two points were sampled, one upstream and one downstream of the Uzein wastewater treatment plant north of Pau (France). The second river selected was the “Riou Mort”, a small river (21.1 km) tributary of the “Lot” (the second longest river in France) in the Massif Central. Due to the presence of the Decazeville coalfield, the “Riou-Mort” river is naturally contaminated with Cd and Zn [35]. Sample analyses are presented in Table 2.

## 5. Conclusions

In this study, the possibility of determining free ion concentrations with the Donnan membrane technique was investigated in order to reduce the measurement time. Comparisons of the concentration determined at Donnan equilibrium (traditional method, 6 h) with that calculated during accumulation showed that the free ion concentration could be estimated with the ISIDORE probe after only 3 h of accumulation for the two metals, Cd and Pb. Comparison between the theoretical and experimental curves showed that Cd and Pb transport was both controlled by diffusion in the membrane. The application of the ISIDORE probe on synthetic and natural river samples showed a good correlation between the two approaches, dynamic and equilibrium, and revealed that special attention should be paid to the Ca concentration when determining the free Cd concentration from the plateau.

## Figures and Tables

**Figure 1 molecules-28-00846-f001:**
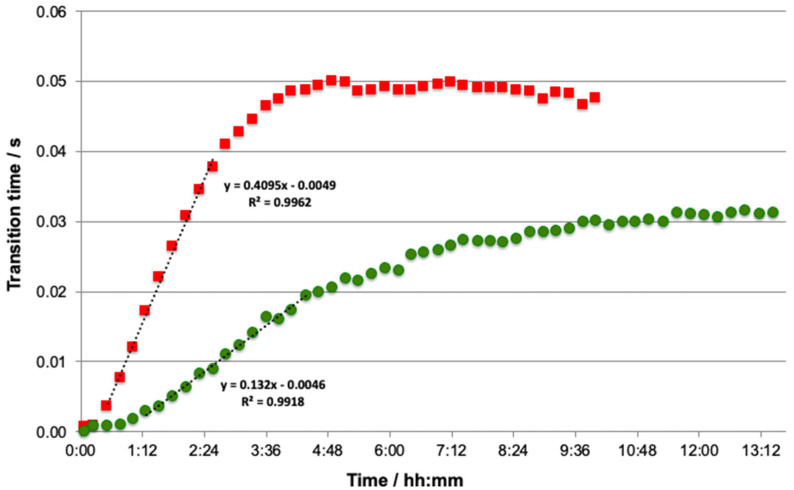
Accumulation curves obtained with the ISIDORE probe with the following total concentrations in donor: Cd at 130 nM (■) and Pb at 72 nM (●). Composition of donor and acceptor solutions: 3 mM Ca(NO_3_)_2_ buffered at pH 4.6 with 5 mM acetate buffer. Analytical conditions: electrodeposition step 120 s at −1.6 V with flux of 2 mL·min^−1^, equilibration time 10 s at −1.6 V without flux, and stripping current *I*_s_ = 10 μA without flux.

**Figure 2 molecules-28-00846-f002:**
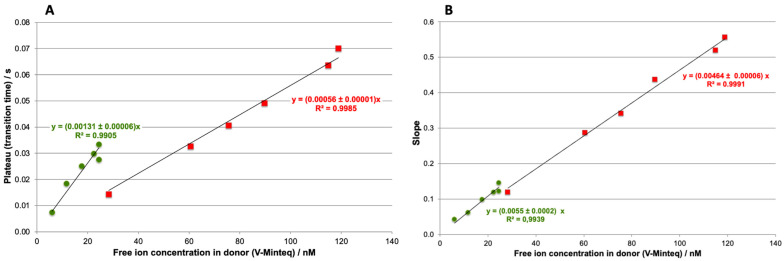
Calibration curves obtained with the ISIDORE probe from the plateau (**A**) and from the slope (**B**) for Cd (**■**) and Pb (**●**). The analytical conditions are the same as in Figure 1.

**Figure 3 molecules-28-00846-f003:**
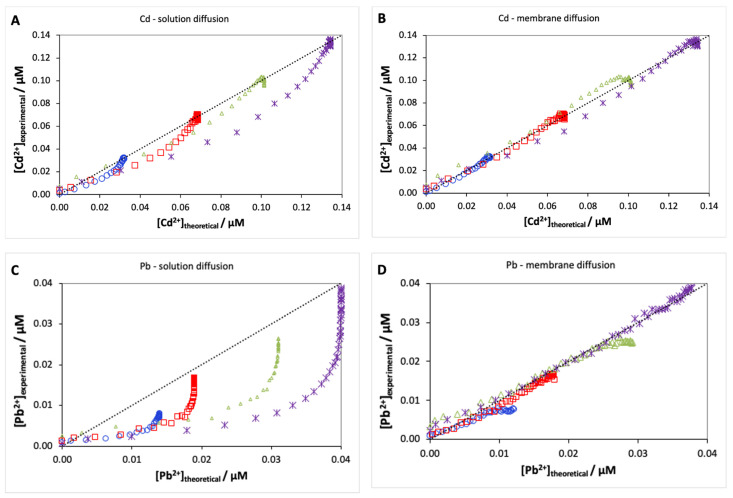
Experimental versus theoretical curves for Cd and Pb when transport is controlled, either by diffusion in the solution (**A**,**C**) or by diffusion in the membrane (**B**,**D**) for 4 free concentrations of Cd (□ 32 nM, ◯ 69 nM, ∆ 101 nM, and ⋆ 134 nM) and Pb (□ 14 nM, ◯ 19 nM, ∆ 31 nM, and ⋆ 40 nM). Experimental points before 30 min were not taken into account as they correspond to the delay required to fill the membrane.

**Figure 4 molecules-28-00846-f004:**
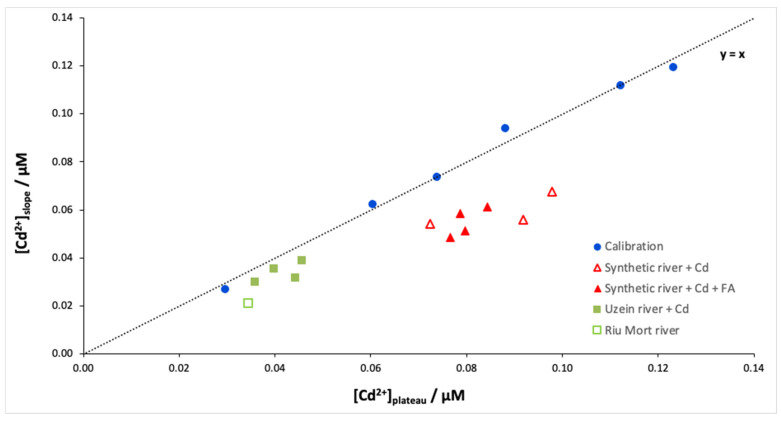
[Cd^2+^]_slope_ versus [Cd^2+^]_plateau_ for the different samples: calibration, Cd-doped synthetic river with fulvic acids at 5 mg/L (4 replicates) or without fulvic acids (3 replicates), and natural samples doped in Cd (Uzein river—4 replicates) or naturally contaminated in Cd (Riu Mort river—1 replicate). Samples were spiked with 89 nM Cd except for the Riu Mort sample, which was naturally contaminated with Cd (36 nM). The analytical conditions are the same as in Figure 1.

**Figure 5 molecules-28-00846-f005:**
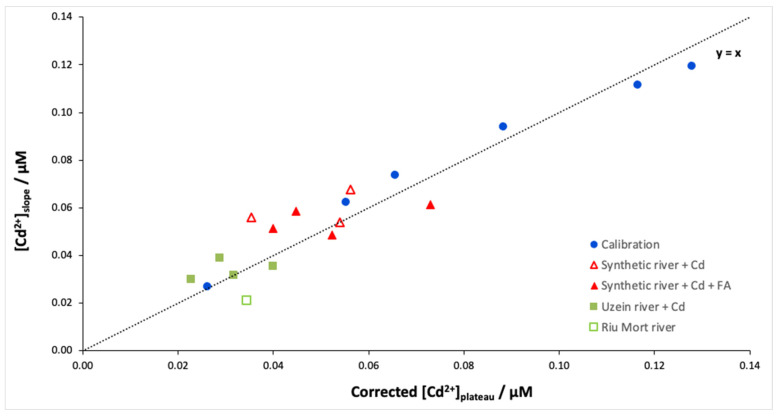
[Cd^2+^]_slope_ versus [Cd^2+^]_plateau_ corrected according to Equation (1) for the different samples: calibration, Cd-doped synthetic river with fulvic acids at 5 mg/L or without fulvic acids, and natural samples doped in Cd (Uzein river) or naturally contaminated in Cd (Riu Mort river). Samples were spiked with 89 nM Cd except for the Riu Mort sample, which was naturally contaminated with Cd (36 nM). The analytical conditions are the same as in Figure 1.

**Figure 6 molecules-28-00846-f006:**
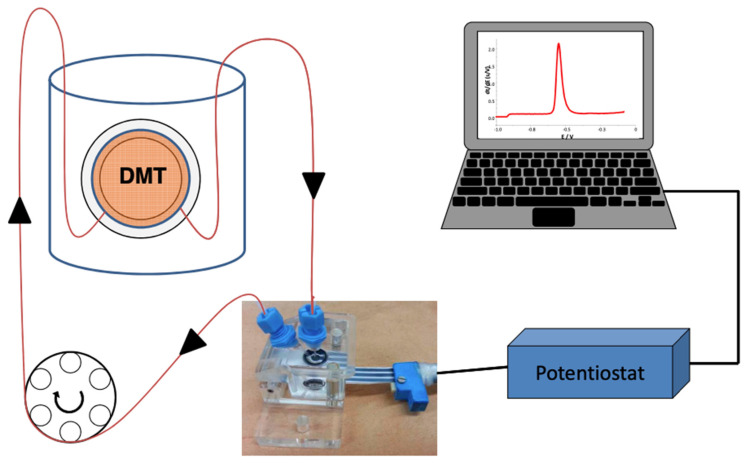
ISIDORE probe: the DMT filled with an acceptor solution of Ca(NO_3_)_2_ is immersed in the donor. The acceptor solution is connected to the flow-cell in which the SPE has been placed [27].

**Table 1 molecules-28-00846-t001:** Free Cd concentrations obtained according to the method plateau or slope.

	Synthetic River		Natural Samples	
	Cd	Cd + FA	Uzein + Cd	Riu Mort
Slope	0.059 ± 0.007	0.055 ± 0.006	0.034 ± 0.004	0.034
Plateau	0.087 ± 0.013	0.080 ± 0.003	0.041 ± 0.004	0.021
Plateau-corrected	0.049 ± 0.011	0.052 ± 0.015	0.031 ± 0.007	0.021

**Table 2 molecules-28-00846-t002:** Composition of natural river samples.

	Riou Mort	Uzein River
pH	7.5	7.44 ± 0.04
Conductivity (μS/cm)	354	285 ± 13
DOC (mg/L)	nd **	1.5
Ca^2+^ (mM) *	3.7	3.2 ± 0.3
Cd (nM)	37	nd ****

* after dopping; ** nd—not detected.

## Data Availability

Not applicable.

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
