# Peer review of "ISIDORE, a Probe for In Situ Trace Metal Speciation Based on the Donnan Membrane Technique and Electrochemical Detection Part 2: Cd and Pb Measurements during the Accumulation Time of the Donnan Membrane Technique"

_molecules, 2023, doi:10.3390/molecules28020846_

Round 1

Reviewer 1 Report

The manuscript entitled " ISIDORE, A Probe for in situ Trace Metal Speciation Based on Donnan Membrane Technique and Electrochemical Detection.Part 2. Cd and Pb Measurements During the Accumulation Time of the Donnan Membrane Technique”. In order to reduce the in situ deployment time, the possibility to measure the concentration of free metal ions (Cd and Pb) while they accumulate in the acceptor solution (dynamic mode) rather than waiting for equilibrium to be reached, is presented.

The topic described is interesting from the perspective of technical analysis and environmental impact. This work is relevant and innovative and may be of interest to the environmental science community.

I recommend its publication after some major revisions

·       Line 77 : please correct ISISDORE probe by ISIDORE

·       line 109 : the slope value was calculated by keeping the points recording during the first three hours corresponding to the linear accumulation of free metals in the DMT

This is not quite what appears on figure 1. For the points in red the linearization starts earlier and for a shorter time than the points in green

·       Line 201 :3.3. Analysis of Synthetic and Natural Samples

Finally, the Isidore probe was applied to synthetic and natural Cd-doped samples.

The authors do not justify their choice to validate their method only on the element cadmium. What about the measurement of lead?

·       Line 206 However, the results of river samples show differences in free Cd concentrations depending on the determination method, slope or plateau.

It would be interesting to specify these differences in a quantitative way to show whether these differences are significant or not. In addition, it would also be necessary to add the uncertainties associated with the concentration measurements

·       Results of analyses of synthetic river doped or not with fulvic acids are presented (figures 4 and 5) without this last parameter ever being taken into account in the discussions! What are the amounts of fulvic acids added in the synthetic waters (concentration of dissolved organic carbon). Do fulvic acids have an influence on the quantification with the used method?

·       Line 225 A good correlation is obtained between the two methods with a correlation coefficient of 0.92.

This sentence is not clear. Please rephrase it: are the results for all types of samples?

·       Line 247 : Stock solutions of Cd, Pb and Hg at 1000 mg L−1 were obtained from Merck.

What is the use of the mercury solution in this study?

·       Line 260: A stock solution of synthetic river water was prepared as follows: 10 mM NaHCO3, 1 mM MgSO4, 2 mM CaCl2·and 0.5 mM KNO3, which corresponds to an ionic strength of 19 mM.

It would be relevant to know the pH of the prepared synthetic river water

·       Line 314 _4.4. Natural Samples

It seems essential to add details on the samples used to validate this technique. What are the compositions of the river waters? Indicate at least the concentrations of dissolved organic carbon, conductivity or ionic strength, the pH and the calcium concentrations.

Are the river waters collected filtered before analysis? If yes, on which type of membrane, which porosity?

Reviewer 2 Report

The authors present a study on the features of ISIDORE, a device based on the Donnan membrane technique for measurement of Cd and Pb in aqueous media. The novelty is limited: the authors apply theory developed by others to their particular device, and do not even modify the equations to be applicable to the ISIDORE dimensions (lines 193-194).

Line 56: what do the authors mean by DMT fits well with WHAM or MINTEQ? These speciation codes are seriously flawed, thus one wonders what connection the authors see here.

The authors state that interpretration of DMT signals is more straightforward than for other chemical speciation techniques. Yet they do not explain how their approach overcomes the difference in transport flux between the calibration and sample media. In line 25 the authors state that the slope of the accumulation – in solutions containing only free metal ions – increases linearly with free ion concentration and thus can be used for calibration. This assumption ignores differences in the mean diffusion coefficient in the presence of labile/partially labile metal complexes.

The authors’ proposal that Cd transport is controlled by solution diffusion and Pb by membrane diffusion is based on measurements in free metal ion only solutions. It is not valid to assume that this holds in the presence of metal complexes, irrespective of their lability. The authors cite Weng et al. 2005 to support their assumption, but Weng et al used Cu complexes with NTA, the lability of which is far lower than most Cd complexes.

Data are shown for Cd-doped samples in Section 3.3. Why was Pb-doping not done?

Reviewer 3 Report

Interesting paper, the Authors want to shorten time necessary to determine concentration of Cd and Pb ions in river samples analysing slope versus plateau of accumulation curves. They have obtained a good correlation, shortening of measurement time but as they have stated – obtained results depend on Ca ions concentration, probably also on pH values and presence of complexing agents and probably presence of other ions like Al3+ in the river samples. So this DMT method, based even with ISIDORE probe still requires further measurements and analysis. Will the Authors succeed ?  - this is still an open question, but this paper shows they are well prepared for that kind of research. In lines 274 there is described procedurę of coating a membrane prior to electrochemical meaurements with thin mercury layer – is it very complicated procedure ? Samll editorial mistakes easy to correct – line 77 should be ISIDORE; line 158 should be m2

Round 2

Reviewer 1 Report

All the comments made have been taken into account, and the modifications and additions have been made. 

Reviewer 2 Report

The authors have made an appropriate re-interpretation of their data. Discussion of the Pb-doping results mentioned in the reply-to-reviewer letter should be added.